# Are ELISA and PCR Discrepancies in the Identification of *Chlamydia pneumoniae* Caused by the Presence of “*Chlamydia*-Related Bacteria”?

**DOI:** 10.3390/microorganisms11010187

**Published:** 2023-01-11

**Authors:** Martina Smolejová, Jana Krčmáriková, Iveta Cihová, Pavol Sulo

**Affiliations:** 1Department of Biochemistry, Faculty of Natural Sciences, Comenius University, Ilkovičova 6, 842 15 Bratislava, Slovakia; 2Synlab Slovakia s.r.o., Limbova 5, 831 01 Bratislava, Slovakia; 3Department of Track and Field, Faculty of Physical Education and Sports, Comenius University, Nábr. Arm. Gen. L. Svobodu, 814 69 Bratislava, Slovakia

**Keywords:** *Chlamydia*, diagnostics, ELISA nested PCR, zoonosis

## Abstract

*Chlamydia* are Gram-negative, intracellular pathogens colonizing the epithelial mucosa. They cause primarily atypical pneumonia and have recently been associated with chronic diseases. Diagnostics rely almost exclusively on serological methods; PCR tests are used rarely because in patients with positive ELISA, it is nearly impossible to identify chlamydial DNA. To understand this issue, we elaborated a reliable and sensitive nested PCR method (panNPCR) for identifying all Chlamydiales species, not only in sputa, but also in clotted blood. Sequencing of the PCR product revealed that 41% of positive sputa samples and 66% of positive blood samples were not infected by *Chlamydia* but with “*Chlamydia*-related bacteria” such as *Rhabdochlamydia* sp., *Parachlamydia* sp., *Protochlamydia* sp., *Neochlamydia* sp., *Mesochlamydia elodeae* and *lacustris*, *Piscichlamydia salmonis*, and *Estrella lausannensis*. Consequently, we propose that there might be more than four human pathogenic *Chlamydia* species. We did not find any clear correlation between increased levels of antibodies and the presence of their DNA. Chlamydialles DNA was found in sputa samples from individuals positive for IgG or IgA but not in blood samples. Thus, elevated IgG and IgA levels are not reliable markers of chronic infection, and the presence of persistent forms should be proved by panNPCR. Apparently, the differences between ELISA and DNA amplification results have three main methodological reasons. The first one is the threshold occurrence of chlamydial genetic material in sputum and blood. The second one is the fact that a significant part of the samples can have DNA with sequences different from those of other species of the order Chlamydiales. The third one is the high background characteristic for ELISA, the absence of paired sera, and the vague interpretation of the gray zone.

## 1. Introduction

*Chlamydia pneumoniae* is a Gram-negative respiratory pathogen responsible for 5% to 40% of lower respiratory tract infections, known as atypical pneumonia or community-acquired pneumonia [1,2]. The figures vary with age, geographical location, the population studied, and, especially, the diagnostic methods used [2,3,4]. However, it is estimated that about 70% of *C. pneumoniae* respiratory tract infections are asymptomatic or with minimal symptoms [4,5,6,7]. This pathogen can also induce a chronic inflammatory response associated with chronic obstructive pulmonary disease (COPD) and asthma [8]. It belongs to the genus *Chlamydia*, which contains obligate intracellular bacteria with a unique biphasic life cycle. Outside of host cells, they exist as infectious, dormant elementary bodies (EBs) that are internalized into host cells by phagocytosis. Inside the host cell, EBs are transformed into metabolically active reticulate bodies (RBs) that replicate in inclusions (modified cellular vacuoles) [9,10]. *Chlamydia* can also persistently infect cells, which can be associated with some chronic diseases. Diagnostics and identification by culture are rare, and immunohistochemical assays, serology, and PCR are laboratory methods almost exclusively used for the diagnosis of acute *C*. *pneumoniae* infection [9,10]. The most commonly used is the evidence for the presence of specific antibodies in blood serum. The microimmunofluorescence (MIF) test and the enzyme-linked immunosorbent assay (ELISA) are currently considered the “gold standards” [2,7,9]. Generally, PCR tests are used rarely because in patients with positive ELISA, it is nearly impossible to identify chlamydial DNA [11,12,13,14,15,16,17,18]. The DNA amplification, specifically real-time PCR, is considered the most promising technology in the development of a rapid, nonculture method, since it works well in the detection of the sibling species *C. trachomatis*, which infects the urogenital tract. For *C*. *pneumoniae*. several dozen in-house PCR assays from clinical specimens such as nasopharyngeal (NPS) or throat swabs, sputa, or pleural fluid have been reported but not validated [9,17,18].

There are several PCR variants, such as real-time PCR, digital droplet PCR, or nested PCR, that help to increase the sensitivity and specificity of detection [19,20,21,22]. However, analysis in silico of the previously published methods revealed their limited use due to the primer design and the extremely high polymorphism of favored target genes such as *ompA* [23]. From the sensitivity and specificity point of view, the target of choice appears to be 16S rRNA, which contains hypervariable regions that are highly specific for a biological species or genus [24,25,26]. Indeed, in a recent “nested” PCR (NPCR) assay, it was possible to specifically amplify the 16S rRNA gene of *Chlamydia* known to infect humans (*C. abortus*, *C. pneumoniae*, *C. psittaci*, and *C. trachomatis*) in several sputum samples. However, sequencing of PCR products revealed that sick individuals with positive *C. pneumoniae*-specific ELISA were infected with other *Chlamydiales* related to Ca. *Rhabdochlamydia porcellionis* and Ca. *Renichlamydia lutjani*, known to infect arthropods [23].

Another drawback is the high prevalence of antibodies against *C. pneumoniae*, especially in the adult population, which is 50% by age 20 and 80% by 60–70 years old [6,9]. In addition, there are extreme differences between serology and DNA amplification assays [12,18,22].

Therefore, the purpose of this work was to elaborate a reliable NPCR test sensitive enough to identify any infection caused by Chlamydiales and compare this outcome with the serological findings.

## 2. Materials and Methods

### 2.1. Patients and Samples

Sputum samples from 10 patients with mostly pneumonia (milder symptoms, specific chest X-ray, no response to beta-lactams) [1,2] were obtained from the HPL s.r.o. Microbiological Laboratory, Bratislava, Slovakia for routine plating tests (four males and six females; mean age: 42; range, 24 to 68 years). Another 156 samples were provided by healthy volunteers from the Faculty of Natural Sciences and the Faculty of Physical Education and Sports [23]. We also examined 95 clotted blood samples (33 men, 62 women, average age 49 years, range 5–90 years) from patients with pneumonia and unclotted blood and sputum samples from one patient with chronic pneumonia, followed for 4 years. The study was approved by the Ethics Committee of the Faculty of Physical Education and Sports under the number 8/2019. The participants provided written informed consent in accordance with the Declaration of Helsinki. The samples were collected in 2018 and 2019, transported to the laboratory, and stored in a freezer at −20 °C. Standards are described in detail elsewhere [23].

### 2.2. DNA Isolation

Sputa. The DNA was obtained from sputa according to Freise et al. [27], with minor modifications as described previously [23].

Blood. Cells from the clotted, and unclotted blood were isolated as described by Xu et al. 2010 [28]. Samples were thawed, and plasma was poured into a new tube. Test tubes with a pellet of precipitated erythrocytes and gel were placed upside down in Falcon tubes and centrifuged for 2 min at 2000× *g*. A portion of the precipitate was cut off with a sterile scalpel and transferred to a plastic column (used for DNA isolation) with a wire sieve, which was placed in a 2 mL Eppendorf tube and centrifuged for 5 min at 14,000× *g*. In this way, we obtained filtered cells (such as erythrocytes and lymphocytes) from which DNA was isolated via several procedures.

Improved methodology for DNA extraction according to Ghatak et al. 2013 [29] and Smolejová et al. 2021 [23]. Briefly, 0.5 mL of erythrocyte slurry or plasma was transferred to a 2 mL plastic tube, and 750 μL of lysis solution (0.1651 M NH_4_Cl; 0.01 M KHCO_3_, 0.001 M tetrasodium EDTA; pH 7.3) was added, following by incubation for 2 min at room temperature. Another 750 μL of lysis solution was added, vortexed for 45 s, and centrifuged for 5 min at 14,000× *g*. The supernatant was discarded, and 1 mL of lysis solution was added to the pellets, briefly vortexed, and centrifuged for 5 min at 14,000× *g*. The supernatant was discarded, and this washing step was repeated two more times. These steps are important for removing red blood cells. The DNA from white blood cells was isolated by the addition of 200 μL of 50 mM NaOH to the pellets. Subsequently, the samples were vortexed, briefly centrifuged, and incubated for 15 min at 95 °C with gentle mixing (600 rpm) on a Thermomixer Comfort (Eppendorf, Hamburg, Germany). In plasma samples, heating was replaced by incubation at room temperature. For neutralization, 32 μL of 1 M Tris-HCl (pH 7) was added to the samples, vortexed, and centrifuged for 1 min at 14,000× *g*. The final pH was 6–7.

AmpliSens^®^ DNA-sorb-B nucleic acid extraction kit (Moscow, Russia). Briefly, 0.3 mL of erythrocyte slurry was transferred to a 2 mL plastic tube, and DNA was isolated according to the manufacturer’s instructions. However, DNA was eluted to 15 μL of 10 mM Tris-HCl, 1 mM EDTA pH 8 (TE) solution. 

QIAamp^®^ DNA Mini Kit (Qiagen, Hilden, Germany). Briefly, 0.1 mL of erythrocyte slurry was transferred to a 2 mL plastic tube, and DNA was isolated according to the manufacturer´s instructions. The DNA was eluted to 50 μL of AE solution. All DNA preparations were stored at −20 °C.

The concentration of isolated DNA was determined with Qubit (QubitTM dsDNA BR Assay) and was for the Ghatak et al. 2013 [29] and Smolejová et al. 2021 [23] procedure (3–15.4 μg/mL), for the DNA-sorb-B nucleic acid extraction kit (˂2–2.36 μg/mL), and for the QIAamp^®^ DNA Mini Kit (˂2–8.62 μg/mL). 

### 2.3. PCR

The primers used in this work are listed in Table 1. The DNA was amplified by FIREpolDNA polymerase FIREPol^®^ DNA Polymerase (Solis BioDyne, Tartu, Estonia) in a 25 μL solution containing 1× buffer B, 0.2 mM dNTPs, 2.5 mM MgCl_2_, 1 pmol/μL of each primer, 2 μL of DNA, and 0.1 U/μL of enzyme. Optionally, the 1× S solution was included if mentioned in the text. The second PCR reaction included the same content, but 5 μL of the DNA from the first reaction was added. To avoid any problem with contamination, two negative controls were introduced after every sample in triplicate experiments. Amplification was performed in a DNA Thermal Cycler (Eppendorf Mastercycler 5330, Eppendorf-Nethel-Hinz GmbH, Hamburg, Germany); the thermal cycles for different primers are listed in Table 2. Real-time PCR was performed using the StepOne™ Real-Time PCR System.

### 2.4. Analysis of PCR Products and Their DNA Sequences

The size of the PCR products was determined by electrophoresis, and PCR products were sequenced and processed as described elsewhere [23] Sequence divergence was determined using the ClustalW program [25], which is part of the CLC Genomics Workbench 9.5 (Qiagen, Hilden, Germany).

### 2.5. ELISA

The ELISA tests for IgA and IgG (Section 3.4) were performed using Anti-*Chlamydia pneumoniae* Human ELISA Kits (Abcam, Cambridge, UK), according to manufacturer´s instructions. IgA, IgG: Cut-off: 10; Gray zone: 9–11; Negative: <9; Positive: >11. Positive and negative serum as well as cut off control serum were included in each experiment. The assay was considered as valid only if the following criteria were fulfilled. Substrate blank: Absorbance value < 0.100; Negative control: Absorbance value < 0.200 and <cut-off; Cut-off control: Absorbance value 0.150–1.300; Positive control: Absorbance value > cut-off. Samples were thought to give a positive signal if the absorbance value was greater than 10% over the cut-off value. Samples with an absorbance value of less than 10% above or below the Cut-off control value should be considered as inconclusive (grey zone), thus neither positive or negative. Samples were considered negative if the absorbance value was lower than 10% below the cut-off. The manufacturer claims 91.7% specificity and 90.2% sensitivity. Cross-reactivity with other Chlamydia was not provided, and the manufacturer does not mention other interferences. 

The ELISA tests for IgA, IgG, and IgM (Appendix A, Section 3.5) were performed with CHLA0510DBNovagnost *Chlamydia Pneumoniae* IgA, CHLG0510DBNovagnost *Chlamydia Pneumoniae* IgG, CHLM0510DBNovagnost *Chlamydia Pneumoniae* IgM kits (Siemens Healthcare Diagnostics Products GmbH, Marburg, Germany), according to the manufacturer´s instructions. IgA, IgM, IgG: Cut-off: 10; Gray zone: 8.5–11.5; Negative: <8.5; Positive: 11.5. Cross reaction with *Chlamydia trachomatis* cannot be excluded. The manufacturer does not mention other interferences. Positive and negative serum, as well as cut off control serum, were included in each experiment, and in each experiment, the test was validated. For an assay to be considered valid, the following criteria had to be met: Negative control Absorbance value < 0.2 and <cut-off; Cut-off control Absorbance value < 0.150 < cut-off < 1.300; Positive control Absorbance value > cut-off. Samples were considered positive if the absorbance value was higher than 15% over cut-off. Samples with an absorbance value up to of 15% above or below the cut-of should not be as clearly positive or negative, and they are classified in gray zone. Samples are considered negative if the absorbance value is lower than 15% below cut-off. A total of 18 samples, 6 positive, 6 negative, and 6 from the gray zone, were also verified by the IgA and IgG by Anti-*Chlamydia pneumoniae* Human ELISA Kits (Abcam, Cambridge, UK), and provided almost the same data. Only two initially barely positive IgG samples had values in the gray zone in this test. Both ELISA tests are routinely used by medical laboratories in Slovakia and are approved in European Union. 

## 3. Results

### 3.1. Pan-Primers

In our previous work, we elaborated a novel NPCR assay allowing the amplification of DNA of *Chlamydia* species accepted as human pathogens. Surprisingly, DNA sequencing revealed that some samples were infected by different Chlamydiales species known as arthropod pathogens. To have a suitable test for the identification of infections caused by any species from the Chlamydiales order, we designed new so-called “pan-primers,” which are complementary to the same regions as we used before [23] but degenerated in several non-conserved positions (Appendix A). In a series of experiments, the conditions for both reactions were optimized, and the detection limit was determined in the same scale (0.34 of cell equivalent in the PCR vial) as reported previously [23].

### 3.2. Chlamydiales in Sputa

In patients with positive ELISA, it is nearly impossible to identify chlamydial DNA [6,7,8,9,10,11], which is associated with low abundance and the low sensitivity of PCR systems [23]. To improve the sensitivity of the reaction, we increased the amount of DNA to 2 μL (4×), and to avoid inhibition of the amplification reaction; 5× more DNA polymerase was added to the first PCR reaction mixture. In addition, the amount of DNA added to the second amplification reaction was increased to 5 μL, and the reaction is referred to as 5 × 2 × 5 panNPCR.

### 3.3. Occurrence of Chlamydiales in Sputum

In the first approach, primers were verified by the more sensitive 5 × 2 × 5 panNPCR reaction on 20 positive sputum samples from the previous work [23]. In 19 cases, chlamydial DNA was amplified, and the origin was confirmed by sequencing. Therefore, we examined DNA from 109 previously negative sputum samples, of which another 25 samples were positive. Identity was confirmed by sequencing, where 10 different haplotypes were identified (Table 3 and Appendix A). Overall, 14 sequences were associated with *C. pneumoniae* and sibling species, but we were not able to unambiguously distinguish them because only short and partially conserved regions were sequenced. The remaining 11 sequences were related to other species from the families “*Chlamydia*-related bacteria,” in particular *Metachlamydia*, *Neochlamydia*, *Parachlamydia*, *Protochlamydia*, and *Rhabdochlamydia* sp. (Table 3, Appendix A).

In summary, out of 148 accepted sputum samples, 61 were identified as positive for the presence of Chlamydiales, resulting in an incidence of up to 41%. Of these positive samples, 41% possessed DNA from “*Chlamydia*-related bacteria”. 

### 3.4. ELISA in Plasma vs. Chlamydial DNA in Sputum

We only had a limited number of samples available to compare the ELISA results from plasma with the presence of chlamydial DNA in sputum. Of 13 examined samples, 12 were positive for IgG and 9 for IgA. In all 13 samples with positive IgA or IgG values, Chlamydiales-specific DNA was present (Table 4).

### 3.5. Chlamydial DNA in Blood

One of our primary goals was to determine whether chlamydial DNA can be proven in individuals with elevated *C. pneumoniae* ELISA antibody levels. If DNA is present, the point is how to interpret elevated values for IgA, IgM, or IgG antibodies. Chlamydial DNA can be determined reliably in sputum, but antibodies are determined in blood or, more precisely, in plasma. It is extremely difficult to obtain a collection of sputum and blood samples with anticoagulants from the same individual. We asked several facilities, but they were able to provide only a limited number of samples (Table 4). Chlamydiales are intracellular parasites, and the presence of their DNA should be preferably determined in blood samples with anticoagulants, where lymphocytes can be separated. However, in diagnostic laboratories, it is considered as inappropriate to take blood twice for clotted blood and with anticoagulants from the point of view of compliance with ethical rules in medicine. Therefore, we had to develop an assay for chlamydial DNA in the same blood samples in which serological tests (ELISA) were performed. At first, we examined several methods of DNA isolation from clotted blood (see Material and Methods section), but the best DNA yield and amplification sensitivity were obtained using a modification of the methods of Ghatak et al. 2013 [29] and Smolejová et al. 2021 [23]. This procedure provides good-quality DNA, which we confirmed by the amplification of human mtDNA as described elsewhere [23]. In all 94 samples, we obtained PCR products with an expected size of 1023 or 250 bp.

### 3.6. ELISA vs. panNPCR in Blood Samples

Since the samples met the quality criteria established for sputum analysis, we examined them for the presence of Chlamydiales DNA using 5 × 2 × 5 panNPCR. Overall, 31 blood samples were positive, and the presence of Chlamydiales DNA was confirmed in 27 samples by sequencing. Data together with ELISA results are summarized in Table 5, and details are provided in Appendix A.

The presence of IgM antibodies without the presence of IgG or IgA antibodies is considered as a sign of primary infection [17,18]. According to our results, this statement is correct, as chlamydial DNA could be amplified from all samples with solely elevated IgM. However, if gray zone values are considered, or samples with simultaneously increased IgG, the number of positives is reduced to 50%. The rise of IgA antibodies is characteristic for reinfections and considered as an indicator of active infection. Our results do not support this opinion because from most of the IgA-positive samples, it was not possible to amplify chlamydial DNA (Table 5). The isolated occurrence of IgG antibodies is believed to be characteristic after an infection had been overcome or for a chronic infection [17,18]. Nevertheless, we were able to prove chlamydial DNA in 6 of 26 samples positive for IgG, which was less than in samples with IgG in the “gray zone” (7/16) (Table 5). Apparently, an increased IgG level is not a reliable marker for chronic infection. However, there is another opinion, namely that consistently present elevated levels of serum IgA antibodies can be used as a marker of chronic infections [17,18,26,31]. Our data do not support this view, as we found Chlamydiales-specific DNA only in one IgA-positive sample (Table 5).

The reliable marker for persistent chlamydia and chronic infection is still a question of debate. Luckily, we also had ELISA test results available, along with blood and sputum samples, from the same patient who had been followed for several years. These samples were subjected to panNPCR analysis, and the results are summarized in Table 6.

This case again indicates that the interpretation of increased IgG levels as a condition after the complete eradication of the pathogen is not correct. Elevated IgG levels together with IgA in the gray zone are a good marker for persistent and chronic infection.

## 4. Discussion

### 4.1. Pan NPCR Reliability

Currently, only four species, namely *C. trachomatis*, *C. pneumoniae*, *C. psittaci*, and *C. abortus*, are recognized as human pathogens [6,10,32,33,34]. As we pointed out previously [23], there are increasing doubts whether only these species play a role in human mucosal disease. In recent years, several new types of *Chlamydia* have been found in human samples from volunteers without disease symptoms [35,36] or patients with respiratory diseases [37,38,39]. In addition, *Candidatus Renichlamydia lutjani* was found in a patient with acute pneumonia, and this infection was recognized by ELISA specific to *C. pneumoniae* [23]. Pan-primers for real-time PCR previously designed to identify the entire genus [39] were not specific enough [23]. Therefore, we designed and elaborated a panNPCR specific for the whole range of *Chlamydiales*, and validated it on sputum and blood samples. The threshold value was as sensitive as that in a previous assay [23]. The origin of any amplified DNA was confirmed by sequencing. Among sputum samples, we encountered the interfering species *Veilonella* sp. in some cases. However, the problem was solved by excluding the S solution from the master mix, without the impact on the sensitivity to detect the other Chlamydiales in truly positive samples. Although the assay is 10 times more sensitive than the only pan test known so far [39], the weak part is the limit of detection in real clinical samples. The actual abundance of target molecules can be estimated by serial dilution of DNA obtained from clinical samples. Unlike synovial fluid samples (in preparation), Chlamydiales-specific DNA cannot be amplified from most of the sputum and all blood samples after 5-fold dilution, indicating the threshold abundance of target molecules. The increased sensitivity is accompanied by a higher risk of contamination. Consequently, due to sensitivity, rules such as those in forensic laboratories should be followed, as described previously [23]. Apparently, this assay is not suitable for routine diagnosis in practice. Therefore, we made an attempt to adjust the panNPCR to one tube and for real-time PCR [40,41,42,43,44]. Unfortunately, this modification provided good sensitivity in dilutions of *C. pneumoniae* TW-183 culture, but in sputum samples, DNA could be amplified only if they were “spiked” with 10× more cells above the detection limit in panNPCR.

### 4.2. Are There Only Four Human Pathogens?

Sequencing of 52 panNPCR products revealed the presence of other genera/species from “*Chlamydia*-related bacteria” in at least 41% sputa of “healthy” probands, from which 41% contained DNA from “*Chlamydia*-related bacteria”. Among blood samples obtained from patients with respiratory diseases, this ratio was even higher (66%), although most of the samples were positive for *C. pneumoniae* antibodies.

Unfortunately, the low mutation/polymorphism rates in 121 bp panNPCR did not allow us to distinguish *C. pneumoniae* from other *Chlamydia* species. Apparently, numerous individuals are not infected with *Chlamydia* species, and we propose that there are more than four human pathogenic *Chlamydia* species.

### 4.3. How to Identify Chlamydial DNA in the Blood

Due to the low abundance, the identification of chlamydial DNA in blood is challenging. Unlike in ELISA, the presence of *C. pneumoniae* is not determined in blood but rather in sputum, nasopharyngeal swabs, and bronchoalveolar lavage fluid, where it can be identified only sporadically (clinical laboratories, personal communication), although several methods have been published [18,45,46]. The assay relies on an improved DNA isolation procedure, and the best results were obtained with a modification of the methods of Ghatak et al. 2013 [29] and Smolejová et al. 2021 [23]. The other two methods, namely the DNA-sorb-B nucleic acid extraction kit and the QIAamp^®^ DNA mini kit, exhibited lower DNA yields and a lower amplification efficiency. The isolations from clotted and unclotted blood in samples taken from one patient collected over time provided the same results (Table 6). We did not find any difference concerning DNA yield and amplification efficiency. The corrections are highlighted in green.

We were able to amplify chlamydial DNA from 66% seropositive blood samples if gray zone values were considered. We tried to improve the method analyzing exome or free-circulating DNA from blood samples [47]. Such DNA consists of extracellular DNA fragments that are released from cells into various body fluids and was first discovered in human plasma. In addition to human DNA fragments, the DNA of various pathogens can also be identified [48]. Moreover, inconsistencies with chlamydial DNA amplification in some samples could be related to DNA degradation. Therefore, we examined plasma samples where free DNA was expected (Appendix A), but DNA was amplified only from a small part of the examined samples. Apparently, freely circulating DNA is not a suitable source for identifying Chlamydiales in blood.

### 4.4. ELISA vs. PanNPCR

*Chlamydia pneumoniae* DNA can be found in only a small part of samples of bronchoalveolar lavage fluid or sputum, even though the patients are diagnosed with pneumonia or have clearly elevated values of individual antibodies in the blood [12,13,14]. Padalko et al. [12] analyzed respiratory tract specimens submitted for *C. pneumoniae* detection collected in four large Belgian hospitals over 2 consecutive years by PCR. Only 0.2% of the 3560 samples were positive for *C. pneumoniae*. Similarly, Miyashita et al. [13] and Noguchi et al. [14] were unable to amplify *C. pneumoniae* DNA from seropositive patients. Up to now, a consensus on how to interpret increased levels of antibodies has been missing. Antibody isotypes are selectively distributed in the body as IgG and IgM, predominantly in the plasma, whereas IgG and monomeric IgA are the major isotypes in the extracellular fluid. Dimeric IgA predominates in secretions across epithelia, and IgE is found mainly associated with mast cells beneath epithelial surfaces. Both IgA and IgE have a much shorter half-life in serum than IgG, which has the longest half-life in serum and can still be present long after an infection was cleared [49]. 

Once *C. pneumoniae* enters the airway, it can infect epithelial cells in the lungs or alveolar macrophages, leading to the inflammatory immune response [50]. When the EB enters the host cell, the inclusion is effective at avoiding the intracellular immune response. However, the pattern recognition receptors NOD1 and NOD2 can detect *C. pneumoniae* and activate an inflammatory response [50]. After initial infection and an innate immune response, the adaptive immune response begins. A coordinated response by CD4^+^ and CD8^+^ T cells is necessary to successfully clear a *C. pneumoniae* infection. The Th1 response is particularly important, as there is evidence that a Th2 response will promote asthma based on elevated immunoglobulin (Ig) IgE titers [50]. The coordinated response by CD4^+^ and CD8^+^ T cells triggers IgM development. At 2 to 3 weeks after a human is first infected by *C. pneumoniae*, the IgM levels will peak, and 2 to 3 months after the infection, they will not be detectable. Furthermore, if there is another infection, IgM is not produced. The IgG levels will peak 6 to 8 weeks following infection, and IgG is produced quickly in any infection after the first one. The IgA is also produced, and its presence, along with that of IgG, can be used to determine if a new infection occurred, especially shortly after an initial infection [50]. Thus, IgM appears within 3 weeks of the first infection, IgG 6–8 weeks, and a ≥4-fold IgG titer rise can be seen upon infection and reinfection, but only elevated values may reflect previous exposure to the organism. Despite considerable efforts, persistent infections are difficult to diagnose, and currently, there are no widely accepted criteria as the IgG antibody can persist long after the acute phase [17,18,26,31,50,51]. The IgA antibodies are naturally short-lived, with a half-life of 5–6 days. Consistently present/elevated IgA antibody titers and elevated serum IgA and IgG antibodies, together with raised C-reactive protein levels, have been proposed as serological markers of persistent *C. pneumoniae* infection. The IgA response could also be a predictor of treatment response [17,18,26,31]. In all patients with elevated levels of *C. pneumoniae*-specific IgG antibodies in the blood, we were able to prove the presence of its genetic material in the sputum. However, a simultaneous increase in IgA occurred in the majority of the samples (Table 4). Although we were able to amplify chlamydial DNA from blood using the more sensitive 5 × 2 × 5 panNPCR, we did not find such a clear association with elevated antibody levels. In many cases, the interpretation was complicated, as the antibody levels were only slightly above the detection threshold (Appendix A). If all types of antibodies were absent, chlamydial DNA could not be amplified from most of the samples (21/28). In cases where IgM, which is produced during infection, was elevated, chlamydial DNA was amplified from most samples. However, this was not the case for IgA, where chlamydial DNA could be confirmed in less than half of the samples. Diagnostics of acute infection based on a single IgG titer should be interpreted with caution. Serum samples obtained from elderly patients and from those with chronic obstructive pulmonary disease have persistently high IgG titers in the absence of a clinically apparent disease response [17,18,26,31]. Nevertheless, an increased IgG level is considered an indicator of an individual’s immune status against specific pathogens. It increases only after a long time from the beginning of the infection, when the pathogen should be eradicated. Most physicians believe that the increase in the IgG titer reflects the condition after the eradication. However, the results obtained from the same patient followed over several years support the second opinion, namely that elevated IgG suggests the permanent presence of persistent forms. In addition, we identified chlamydial DNA in almost a quarter of the IgG-positive samples and in almost half of the samples where the IgG levels were in the gray zone. There are cases where the level of antibodies did not increase but chlamydial DNA was present. The most plausible explanation for this is the infection with related *Chlamydia* species; consequently, the antibodies may not have had sufficient affinity and avidity. Apparently, the differences between ELISA and PCR, emphasized previously [2,4,7,9,12,13,14], have two main reasons. The most profound ones are the threshold values of chlamydia genetic material in sputum and the degradation rate [23]. The lower contribution is associated with the presence of sequentially different DNA from other species of the order Chlamydiales, which can be the cause of positive ELISA tests. Therefore, when interpreting the ELISA results, it is important whether the increase in IgG is a result of the post-eradication state or the outcome of the permanent presence of persistent forms. Apparently, a simultaneous increase in IgG and IgA is a suitable but not absolute marker for the persistent presence of Chlamydiales. However, if the chronic state continues and the level of antibodies does not decrease, the presence of Chlamydiales should be proven by panNPCR amplification. Apparently, further progress in this field will result in the improvement of ELISA and DNA amplification methods, especially regarding the detection limits.

## 5. Conclusions

We elaborated a reliable and sensitive nested PCR method for identifying Chlamydiales species, not only in sputa, but also in clotted blood, where a specific DNA isolation procedure is required. The method is as sensitive as NPCR designed to detect DNA from all *Chlamydia* species recognized as human pathogens [23]. Sequencing of the PCR product revealed that 41% of positive sputa and 66% of positive blood samples were not infected by *Chlamydia* but by “*Chlamydia*-related bacteria”. Therefore, the spectrum of chlamydial human pathogens needs to be expanded and revised. We did not find any clear correlation between elevated levels of antibodies and the presence of specific DNA. Although Chlamydiales DNA was found in sputa samples from individuals positive for IgG or IgA, but not in blood samples, therefore, elevated IgG and IgA levels are not reliable markers for chronic infection, and the presence of persistent forms should be proved by sufficiently sensitive DNA amplification method such as panNPCR. Apparently, the differences between the ELISA and DNA amplification results are associated with (i) the threshold occurrence of chlamydial genetic material in sputum and blood, (ii) the presence of DNA with sequences different from those of other Chlamydiales species, and (iii) the limitations of ELISA (high background, absence of paired sera, and vague interpretation of the gray zone).

## Figures and Tables

**Table 1 microorganisms-11-00187-t001:** PCR primers used in this study.

Primer	Sequence 5′→3′	Target/Size	T_m_ [°C]
Panout	RYGGRGAAARNGGAATTCCA	16S rDNA 218 bp	54.6
Pshort down	YATACTTAACGCGTTAGCTMCRACAC		55.4
Panin	GTGGCGAAGGCGCTTTTC	16S rDNA 126 bp	55.7
PChtin	GGTTGAGWCYRNYYACAYCAAGT		54.3
MT for	CACCATTAGCACCCAAAGCT	mtDNA 1023 bp	51,9
MT rev	CTGTTAAAAGTGCATACCGCCA		54,6
16S1 F	CCCGCCTGTTTACCAAAAACAT	mtDNA 250bp	56.8
16S1 R	AAGCTCCATAGGGTCTTCTCGTC		54.7

**Table 2 microorganisms-11-00187-t002:** PCR programs applied in this study.

Primers	Program
Panout/Pshort down	94 °C—3 min, 35 × (94 °C—45 s, 54 °C—1 min, 72 °C—1 min), 72 °C—5 min, 14 °C
Panin/PChtin Chp down	94 °C—3 min, 30 × (94 °C—45 s, 54 °C—1 min, 72 °C—1 min), 72 °C—5 min, 14 °C
MT for/MT rev	94 °C—5 min., 30 × (94 °C—1 min, 54 °C—1 min, 72 °C—1 min), 72 °C—3 min, 14 °C
16S1 F/16S1 R	94 °C—5 min., 30 × (94 °C—1 min, 55 °C—1 min, 72 °C—1 min), 72 °C—3 min, 14 °C

**Table 3 microorganisms-11-00187-t003:** Haplotype summary.

Haplotype	Number of Samples	Polymorphisms	Species
1	12		*C. pneumoniae* *
2	1	46T→G	*C. pneumoniae*
3	1	34T→Y	*C. pneumoniae*
4	1	13A→G; 41T→G	*Parachlamydia*, *Rhabdochlamydia*
5	1	11G→T; 51G→T	*Protochlamydia*
6	1	11G→T; 51G→C	*Protochlamydia*, *Neochlamydia*
7	1	11G→T; 44C→T	*Protochlamydia*, *Neochlamydia*
8	1	11G→T	*Metachlamydia lacustris*
9	4	55C→T	*Parachlamydia*, *Rhabdochlamydia*
10	2	5A→T; 40T→A; 55C→T	*Rhabdochlamydia*

* reference sequence *C*. *pneumoniae* AR39 (NC_002179.2) [30]; sequenced 58 nt region.

**Table 4 microorganisms-11-00187-t004:** ELISA in plasma vs. panNPCR in sputa.

Number	Sample	IgA (Cut off > 11)	Result	IgG (Cut off > 11)	Result	LNPCR ^a^ 461 bp	panNPCR 121 bp
1	1080	27.7	+	21.5	+	+ ^CRB^	+ ^CRB^
2	1082	7.75	−	24.7	+	−	+
3	1089	32.1	+	38.8	+	−	+
4	1096	6.49	-	33.6	+	−	+
5	1779	26.6	+	67.1	+	−	+
6	1788	26.5	+	72.2	+	−	+
7	1803	6.59	−	16.8	+	−	+
8	1807	34.1	+	67.4	+	−	+
9	1828	14.4	+	˂10	−	−	+
10	1838	22.0	+	62.1	+	−	+
11	DS	23.6	+	73.4	+	+	+
12	PS	11.7	+	25.5	+	+	+
13	45SP	10.2	Gray zone	18.6	+	+	+

^a^ from reference [23]. IgA, IgG: Cut-off: 10; Gray zone: 9–11; − negative: <9; + positive: >11 ^CRB^ DNA from *Ca. Renichlamydia lutjani*.

**Table 5 microorganisms-11-00187-t005:** ELISA vs. panNPCR in blood—summary.

	Number of Samples	IgM	IgG	IgA	panNPCR
1	0	+	+	+	+
2	21	-	-	-	-
3	7	-	-	-	+4 ^crb^
4	3	+	-	-	+2 ^crb^
5	3	+/-	-	-	+1 ^crb^
6	0	+	+	-	+
7	2	+	+/-	-	+2 ^crb/1^*
8	1	+	+	-	-
9	6	+/-	-	-	- ^1^*
10	2	+/-	+	-	+2 ^crb^
11	1	+/-	+/-	-	+1 ^crb/^
12	2	-	+	-	+1 ^crb^
13	16	-	+	-	-
14	0	-	+	-	+
15	9	-	+/-	-	-
16	6	-	+/-	-	+3 ^crb^
17	5	-	+	+	-
18	2	-	+/-	+	+ ^1^*
19	1	-	+	+/-	-
20	1	-	+/-	+	-
21	1	-	+/-	+/-	-
22	1	-	-	+	+1 ^crb^
23	4	-	-	+/-	-

+ positive; - negative; +/- gray zone; ^1^* amplified from plasma, ^crb^ DNA from *Ca. Renichlamydia lutjani*.

**Table 6 microorganisms-11-00187-t006:** ELISA vs. panNPCR from the sputa and blood from the same individual.

Sample	Date	Age	IgG	IgA	panNPCR	
					Sputa	Blood
1′	17 October 2013	56	+52.8/cut off 15	+/-9.39/cut off 12	+	+
1	17 October 2013	56	+35.5/cut off 11	+/-8.5/cut off 11	+	+
2	21 November 2014	57	+25.5/cut off 11	+11.7/cut off 11	+	+
3	30 July 2015	58	+34.2/cut off 11	+/-10.5/cut off 11	+	+
4	30 September 2017	60	+22.3/cut off 11	+/-9.8/cut off 11	+	+

+ positive; - negative; +/- gray zone; sample 1′ unknown ELISA provider; samples 1–4 data from Anti-*Chlamydia* pneumoniae Human ELISA IgA and IgG Kits (Abcam, Cambridge, UK).

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
