# Peer review of "Are ELISA and PCR Discrepancies in the Identification of Chlamydia pneumoniae Caused by the Presence of “Chlamydia-Related Bacteria”?"

_microorganisms, 2023, doi:10.3390/microorganisms11010187_

Round 1

Reviewer 1 Report (Previous Reviewer 2)

From my point of view the manuscript is ready for pubblication, and ELISA tests appear correctly executed and evaluated.

Reviewer 2 Report (Previous Reviewer 1)

Can be accepted in present form

This manuscript is a resubmission of an earlier submission. The following is a list of the peer review reports and author responses from that submission.

Round 1

Reviewer 1 Report

Authors have put commendable effort to develop a reliable and sensitive nested PCR method for identifying all Chlamydiales species not only in sputa but also in clothed blood. I have few comments.    In Table 5,  1. In row 3, no elevation of either of the antibodies (IgM, IgG and IgA), but still there is amplification of DNA from Ca. Renichlamydia lutjani. If possible, please explain.   2. In row 12, IgG is elevated and there is amplification of DNA from Ca. Renichlamydia lutjaniIn row 13, IgG is elevated in all 16 samples, but there is no amplification. Please explain.    3. Please define the grey zone clearly.    Minor comments -  English language must be taken care of. Few corrections are given below. The English language must be thoroughly checked in the manuscript.    - Line 47, 'Inside the host cell EB'.... correction- comma must be placed after 'cell'   - Line 20-21, Sequencing of PCR product revealed that 41% of positive sputa samples and 66% of positive 20 blood samples are not infected by Chlamydia but with chlamydia related bacteria  correction - and (in red) should be added   - Line 43, 'It has also has the ability' Correction - It also has the ability

Author Response

Reviewer 1

Comments and Suggestions for Authors

Authors have put commendable effort to develop a reliable and sensitive nested PCR method for identifying all Chlamydiales species not only in sputa but also in clothed blood. I have few comments.    In Table 5, 

  1. In row 3, no elevation of either of the antibodies (IgM, IgG and IgA), but still there is amplification of DNA from Ca. Renichlamydia lutjani. If possible, please explain.

This we explained in the text as a results of low antibody cross-reactivity. Thus antibodies against C. pneumoniae may not have sufficient avidity against Chlamydia-related bacteria.

  1. In row 12, IgG is elevated and there is amplification of DNA from Ca. Renichlamydia lutjani. In row 13, IgG is elevated in all 16 samples, but there is no amplification. Please explain.

The same explanation. Results depend on the cross-reactivity and sequence differences. Sometimes the differences between the ELISA and DNA amplification results are associated with the threshold occurrence of chlamydial genetic material in sputum and blood.

  1. Please define the grey zone clearly.

The gray zone is mentioned in the section 2.5 and the source are instruction manuals.

    Minor comments -  English language must be taken care of. Few corrections are given below. The English language must be thoroughly checked in the manuscript.    - Line 47, 'Inside the host cell EB'.... correction- comma must be placed after 'cell'   - Line 20-21, Sequencing of PCR product revealed that 41% of positive sputa samples and 66% of positive 20 blood samples are not infected by Chlamydia but with chlamydia related bacteria  correction - and (in red) should be added   - Line 43, 'It has also has the ability' Correction - It also has the ability

We thank the reviewer for the constructive comments. We fixed most of the parts according to the suggestions. New version of our MS was corrected by professional language editor. The corrections are highlighted in green.

Reviewer 2 Report

Authors present a PCR method for identifying all Chlamydiales specie in sputo and clotted blood, comparing the results with antibody detection in ELISA: The manuscript is exhaustive, despite sometimes present poor readability. I suggest the following major and minor points in order to improve the manuscript.

-manuscript title: I propose to modify with one of the following “Are ELISA and PCR discrepancies in the identification…” or “Are ELISA vs PCR discrepancies in the identification…”  

-please full explain the panNPCR acronym at least the first time that it appears in the text

-please substitute “clothed blood” with “clotted blood” into the abstract and in the rest of the manuscript.

-please insert the corresponding updated literature supporting comments on lines 46-49. In general, among the manuscript some references should be refresh, several recent publications can be introduced. In example, in the discussion section (lines 257-259) can be inserted fresh references regarding chlamydia infections and SARS-COV2.

-Introduction lines 75-77 should be re-written. Described percentages of antibodies are nor clear among groups. Moreover, references 23 and 24 are not correct, they do not describe percentage of antibodies. I strongly suggest revising all references because there are a high percentage that do not correspond with the text.

-Materials and methods, the number of the ethical committee approval is lack, please insert the univocal number.

-materials and methods line 106. I suggest to modify title “Laboratory modification of Ghatak et al. 2013 (27) and Smolejová et al. 2021(20)” with a title like “Improved methodology for DNA extraction” or similar.

-Lines 150-155. I suggest the removal of references to webpage of manufacture instructions. It should be enough explain that the instructions have been followed. Morevorer, reference numbers are incorrect again. Please check carefully all references.

-Tables and Figures has not been numbered in order of appearance. Please check tables and figure numbers, which should be numbered in order of appearance. Please also uniform the names, sometimes appears as “Figure” other as “Fig”. The same for tables.

-Lines 188-190. Authors claim “61 were identified as positive for the presence of Chlamydiales giving the incidence of up to 41%, from which 41% comes from "Chlamydia-related bacteria". It is not clear, author means that 41% of 61 patients are chlamydia related bacterial? Please clarify this point.

-Section 3.4. please add one comment regarding IgA and IgG differences in samples. Not all patients are positivity to IgA and IgG, please specify the number out of positives for each group.

-Line 202. Please remove “(20 and this work)”

-Lines 230 and 280. I suggest to use a moderate language modifying “this rule is false” or “this idea is false” with moderate expression like in example“our results do not support this hypothesis” or “our results clarify that…”

Minor revisions:

-Abstract line 20: “…41% of positive sputa samples 66% of positive…” may be authors intended to writhe “41% of positive sputa samples AND 66% of positive...”

-Introduction line 43 “It has also has the ability…” with “It also has the ability”

-Line 97 “minor modifications described previously in (20).” With minor modifications previously described (20)”

-Line 122 please insert a space in between “to” and “15”

-Line 181 “we were not able TO univocally.”

-Line 222 Please remove the “,”

-Line 321 please insert a space in “3 week”.

Author Response

We thank the reviewer for the constructive comments. We fixed most of the parts according to the suggestions. We tried very hard to update the literature, but it didn't work out better. Some of your suggestions have been corrected by the language editor. The corrections are highlighted with green in the second corrected version.

Reviewer 3 Report

Manuscript ID: microorganisms-2078262

Review report on the article entitledAre ELISA PCR discrepancies in the identification of Chlamydia pneumoniae caused by the presence of "Chlamydia-related bacteria"?

This article describes a sensitive nested PCR method for detecting and identifying various species in the order Chlamydiales in sputa and clotted blood. ELISA and PCR/sequencing results were compared. The study did not find a clear correlation between elevated level of antibodies (ELISA) and the presence of bacteria (PCR amplification of target genes and sequencing).

However, it appeared that the contribution of the humoral immune response-dependent factors, bacteria-dependent factors and host-dependent factors in the discrepancies between the ELISA and PCR were not taken into consideration by the authors. For example, antibody isotypes are selectively distributed in the body (e.g. IgG and IgM predominate in plasma, whereas IgG and monomeric IgA are the major isotypes in extracellular fluid within the body. Dimeric IgA predominates in secretions across epithelia. IgE is found mainly associated with mast cells just beneath epithelial surfaces). Both IgA and IgE (extremely short) have a much shorter half-life in serum (or plasma) than IgG, but this does not necessarily mean that IgA and IgE are not present in the host, they can be in different locations. IgG has the longest half-life in serum and can still be present long after an infection was cleared. The presence of IgG is therefore not always an indicator of an ongoing acute infection nor a chronic infection. In addition, independent of reactivation via poorly understood mechanisms; resident long-lived plasma cells (memory) secrete antibodies sporadically for months, years or a lifetime. The specific locations of antibodies at any given time together with the type of sample used for detection will have a direct effect on the results. Similarly, the bacteria will be intracellularly and/or extracellularly in circulation and/or target organs at several different time points, which may result in high to zero bacterial yields depending on the sample used at any given time. Obtain review articles about the humoral immune response that include the kinetics and major locations of isotype switched antibodies, as well as about the intracellular/extracellular locations of the bacteria and kinetics of the bacterial life cycle.

It is understandable that the authors had to develop an assay for the isolation of lymphocytes from clotted blood samples due to the ethical constrains. However, since blood samples with anticoagulants are required for the effective isolation of lymphocytes, it is important to include comparative data from both methods in this study. If this was not done, it should be included as a limitation of this study.

While this article showed the need to develop additional assays to be used in conjunction with the ELISA, the lack of controls in this study is very problematic. It is essential to include results of positive and negative controls in all experiments. The authors must provide the results of positive and negative controls to validate their observations before this work can be published. This include positive and negative controls for the human samples as well as the assays used. Without clearly showing that non-specific binding, etc did not occur, the results of the ELISA cannot be conclusively linked to antigen/antibody specific Chlamydia or Chlamydia-related bacteria that were identified by PCR and sequencing.

Specific comments:

Abstract: Check if the spelling of the bacterial names are correct and if the bacterial names should be italicized or not.

Abstract, Lines 23-24: “Apparently, the opinion about four chlamydial species as human pathogen should be revised” - This is the opinion of the authors based on their interpretation of the results. It is not acceptable to use these type of phrases or statements in scientific writings. If the data of a study is contradictory to other studies then the authors must clearly state that and use terms like ‘we speculate’, ‘we propose’, etc. (For example, ‘based on the results of this study, our group propose that there might be more than four human pathogenic chlamydial species’). There are several instances in the article where these bold claims were made. Rewrite here and throughout the entire article.

1. Introduction, Line 42: C. pneumonia - Do not use abbreviations without writing out the whole name first. Rewrite line 38: Chlamydia pneumonia (C. pneumonia). Correct here and elsewhere.

1. Introduction, Line 45: “It belongs to Chlamydiae that are”- Specify that this is the genus. Avoid using various bacterial names without stating the specific classifications. Include the bacterial taxonomy (e.g. C. pneumonia, a member of the genus Chlamydiae, family Chlamydiaceae in the order Chlamydiales).

1. Introduction, Line 49: “Chlamydiae”- Chlamydiae should be italicized. Check if the bacterial names should be italicized or not. Change here and throughout the article.

Introduction, Line 70: “amplify DNA of Chlamydia” - Use the correct scientific terminology [e.g.  amplify the target DNA sequence(s), or gene(s), or segment(s) of DNA]. Change here and throughout the article.

1. Introduction, Line 77: “respectively (40–90%)”- Both the two different age groups and their respective percentages were provided; it is therefore unclear what the meaning of respectively (40–90%) is?

2.1. Patients and samples, Line 84: “mostly atypical” - Explain mostly atypical.

2.1. Patients and samples, Lines 86-87: Provide information about the disease status, the number of volunteers and list the type of samples obtained.

2.1. Patients and samples, Line 88: “clothed blood samples”- Should be clotted blood, correct here and throughout the article.

2.1. Patients and samples, Line 88: Provide information about the disease status of the patients.

2.1. Patients and samples, Lines 84-94: Include information about the negative and positive control groups.

2.4. Analysis of PCR products and their DNA sequences, Line 146: “as described in (20)” - Change to ‘as described by reference (20).’ Make sure to use the correct reference styles throughout the article.

2.4. Analysis of PCR products and their DNA sequences, Line 147: “CLC Genomics Workbench 9.0 program” - Include a reference (website). Also, check the whole article that all references are included.

2.5. ELISA, Line 147: Include the grey zone parameters used in this study. Provide the positive and negative controls used for non-specific binding, cross-reactivity, etc.

3.4. ELISA in plasma vs Chlamydial DNA in sputum, Line 194: “(Tab 4)”- Should be Table 4, correct here and elsewhere. Additionally, confirm that all the figures and tables used in the article or supplementary material are included and referred to (Table S1 is not referred to in the article).

Table 4. ELISA vs panNPCR in sputa, Line 195: Change to ELISA in plasma vs panNPCR in sputa.

3.5. Chlamydial DNA in blood, Lines 199-217: Include the following information, (1) The presence/absence of EBs in blood, are EB,s circulating freely or associated with cells (provide the specific cell type(s). (2) Include the specific cell subset(s) that RBs are detected in during bacteremia.

3.6. ELISA versus panNPCR in blood samples: All of the ‘factual information’ about antibodies were obtained (references used) from either Chlamydia- review or experimental articles. This must be clearly indicated in this article. It currently seems that the authors discussed antibody responses in general, which in turn resulted in multiple false information about antibodies. For example, 3.6. ELISA versus panNPCR in blood samples, Lines 228-230: “The rise of IgA antibodies is characteristic for reinfections and is considered as an indicator of active infection” - This statement is incorrect; all antigen-specific, high-affinity, isotype switched antibodies (IgA, IgG and IgE) are produced/secreted in the germinal centres during a primary immune response. All of the antigen-specific, high-affinity, isotype switched antibodies (IgA, IgG and IgE) will be secreted during subsequent reinfections (memory immune responses). However, if high levels of IgA is a characteristic feature of Chlamydia reinfections and considered an indicator of an active Chlamydia infection, it should be clarified. Make sure to directly link all the referred antibody information with Chlamydia. Rephrase here and throughout the article.

3.6. ELISA versus panNPCR in blood samples, Lines 230-231: “Outcome from our data indicates that this rule is false because from most of IgA positive samples it is was not possible to amplify chlamydial DNA (Table 5)” - See previous comment (Abstract) about making these bold claims. Rewrite here and elsewhere.

Author Response

We thank the reviewer for the constructive comments. We fixed most of the parts according to the suggestions. But some were beyond our technical capabilities, such as the presence/absence of EBs in blood. None of the patients had bacteremia. Because we cannot amplify longer amplicons from blood samples, we believe that degraded Chlamydial DNA is mainly present as well as in the sputum. It occurs mostly in the cell fraction, although it can also be amplified sporadically from plasma fractions. We apologize for the first version of the article, which was not corrected and one of the raw versions was mistakenly uploaded. Nevertheless, the new version of our MS was corrected by professional language editor. We compared the isolations from clotted and un-clotted blood only in samples taken from one patient collected over time. We did not find any difference concerning DNA yield and amplification efficiency. The corrections are highlighted in green.

Round 2

Reviewer 2 Report

The manuscript has been sensibly modified. Despite I can see only word corrections I understand that the major points have been solved. 

Reviewer 3 Report

Manuscript ID: microorganisms-2078262

Review report 2 on the article entitledAre ELISA PCR discrepancies in the identification of Chlamydia pneumoniae caused by the presence of "Chlamydia-related bacteria"?

It was clearly stated in review report 1 that the authors must provide the results of positive and negative controls to validate their observations before this work can be published (see below). In addition to not addressing the lack of controls in their letter, the authors have not included the results of the requested controls in the article.

 While this article showed the need to develop additional assays to be used in conjunction with the ELISA, the lack of controls in this study is very problematic. It is essential to include results of positive and negative controls in all experiments. The authors must provide the results of positive and negative controls to validate their observations before this work can be published. This include positive and negative controls for the human samples as well as the assays used. Without clearly showing that non-specific binding, etc did not occur, the results of the ELISA cannot be conclusively linked to antigen/antibody specific Chlamydia or Chlamydia-related bacteria that were identified by PCR and sequencing.”